# Numerical Simulation of Tidal Current and Sediment Movement in the Sea Area near Weifang Port

**Jiarui Qi** [1,2], **Yige Jing** [2], **Chao Chen** [1] and **Jinfeng Zhang** [2,*]

1 College of Harbour and Coastal Engineering, Jimei University, Xiamen 361021, China; qi_jiarui@163.com (J.Q.); chenchaojmu@126.com (C.C.)
2 State Key Laboratory of Hydraulic Engineering Simulation and Safety, Tianjin University, Tianjin 300072, China; jjgg@tju.edu.cn
* Correspondence: jfzhang@tju.edu.cn

**Abstract:** This paper uses the finite-volume community ocean model (FVCOM) coupled with the simulating waves nearshore (SWAN) in a wave–current–sediment model to simulate the tidal current field, wave field, and suspended sediment concentration (SSC) field in the sea area near Weifang Port, China. The three-dimensional water-and-sediment model was modified by introducing a sediment-settling-velocity formula that considers the effect of gradation. Next, the SSCs calculated by the original and modified models were compared with the measured data. The SSCs calculated by the modified model were closer to the measured data, as evidenced by the smaller mean relative error and root-mean-square error. The results show that the modified coupled wave–current–sediment model can reasonably describe the hydrodynamic characteristics and sediment movement in the sea area near Weifang Port, and the nearshore SSCs calculated by the modified model were higher than those calculated by the original model.

**Keywords:** nonuniform sediment; sediment settling velocity; sediment transport; coupled numerical model; Weifang Port

## 1. Introduction

China's Weifang Port is located on a typical silty coast. In response to the actions of wind and waves, the sediment content in the waters near a silty coast increases greatly, and a high proportion of this increase is suspended load, so channel siltation is prone to occur. Therefore, the accurate calculation of the hydrodynamic and suspended sediment conditions in response to the actions of wind and waves is necessary to obtain correct channel erosion and deposition results.

Numerical simulations of sediment movement have been developed and improved over the years, and many estuarine and coastal sediment-calculation models have been designed. An estuarine and coastal sediment model is a system integrating a hydrodynamic model, a wave model, a sediment model, and a terrain-evolution model. Specifically, the hydrodynamic model and the wave model are used to simulate the current movement and wave evolution in large-scale sea areas, and the sediment model and terrain-evolution model are used to simulate the transfer patterns and the erosion and deposition characteristics of suspended loads and bed loads in sea areas with hydrodynamic environments. The most widely used hydrodynamic-sediment models include ROMS [1], Delft3D [2], ECMSED [3], TELEMAC [4], SCHISM [5], and FVCOM [6], and the most popular wave models include SWAN [7] and WAVEWATCH [8]. The early models were mostly two-dimensional planar models. With the improvement of computer technology and performance, three-dimensional models have become mature. Since waves and currents coexist in actual estuarine and coastal environments, coupled wave–current–sediment models have been developed and widely applied in engineering practice. Wang [9] established a three-dimensional, unstructured, fully coupled wave–current numerical model. Yang [10]

established a dynamically coupled wave–hydrodynamic model, the finite-volume community ocean model (FVCOM) coupled with the simulating waves nearshore (SWAN) with a model-coupling toolkit (MCT) coupler. Dietrich et al. [11] constructed a coupled SWAN-ADCIRC model by integrating the unstructured-mesh SWAN spectral-wave model and the ADCIRC shallow-water model. Warner et al. [1] established a coupled ROMS-SWAN model and used it to simulate the sediment movement in Massachusetts Bay during storm surges. Luo [12] numerically simulated the water and sediment transport and long-term topographic evolution in Liverpool Bay, UK, by reorganizing the tidal-current module of TELEMAC, the wave module of TOMAWAC, and the sediment module of SISPHE.

As the basic problem in sediment dynamics, the sediment-settling velocity cannot be ignored in sediment-calculation models. In actual estuarine and coastal environments, natural sediments exist in the form of mixed sediments with nonuniform particle sizes, but most of the settling-velocity formulas are designed for uniform sediments. Therefore, the median particle size of the sediment is usually substituted into the sediment-calculation formula when numerically simulating sediment erosion, deposition, transport, etc. [13,14]. The formulas proposed by Stokes [15], Oseen [16], and Krone [17] for calculating the settling velocities of spherical sediments in still water are commonly used. When considering the influence of the irregular shapes, surface roughness, and physical composition of natural sediments, some scholars prefer to use the formulas proposed by van Rijn [18], Soulsby et al. [19], and Cheng [20] to calculate the settling velocities of natural sediments. Regarding the restricting effect of the sediment concentration on the sediment-settling velocity, some scholars favor the constrained settling-velocity formulas proposed by Richardson and Zaki [21], Camenen [22], and Slaa et al. [23]. Fang et al. [24] adopted the summation $\sum_{k=1}^{n} P_{ok}\omega_k$, where $P_{ok}$ is the percentage of sediments with particle size $d_k$, and $\omega_k$ denotes the settling velocity of the $k$th sediment component in still water, to calculate the mean settling velocity of nonuniform sediment. They obtained a different transport-capacity equation from that of the uniform sediment, which indicated that there were differences in the calculation of the transport process between the nonuniform sediment and the uniform sediment. To account for the effect of nonuniform sediment, Molinas et al. [25] and Wu et al. [26] proposed a variable, representative particle size for nonuniform sediment-transport calculations. Smart and Jaeggi [27] proposed a nonuniformity factor expressed by $d_{90}/d_{30}$ to explain the effect of the particle-size distribution. Shen and Rao [28] adopted $G = 0.5(D_{84}/D + D_{50}/D_{16})$ as a size-gradation factor. Sun et al. [29] used the functional relationship between the settling velocity of a single particle and the relative diameter and geometric standard deviation of nonuniform sediment when calculating the SSCs in a vertical profile. The reasonable description of the settlement of nonuniform sediment is also important for the study of sediment-transport capability and of current patterns.

Few mathematical models consider the characteristics of sediment movement in waters near silty coasts in response to the combined actions of waves and currents, and the influence of sediment gradation is not considered in single-component models. Based on the FVCOM-SWAN coupled wave–current–sediment model, this paper reports the simulation of a hydrodynamic environment and suspended-sediment movement in the sea area near China's Weifang Port in response to the combined actions of waves and currents. According to previous studies [24,29], the calculation method for the settling velocity of non-uniform sand is different from that used for uniform sand, which affects the simulation results of sediment transport. However, silty coast sediments often have a strong sorting ability, and the effect of the gradation on the average sediment-settling velocity cannot be ignored. The formula for sediment-settling velocity commonly used in mathematical models cannot fully reflect the gradation characteristics of mixed sediments, and the method in which the mean sediment-settling velocity is calculated by substituting the median particle size of the sediment may be overly simplified. Therefore, the model in this paper describes the settlement process of nonuniform sediment by introducing a sediment-settling-velocity formula with a coefficient that considers the effect of gradation.

Furthermore, this paper explores the difference between simulated sediment-content results before and after considering gradation.

## 2. Numerical Models

### 2.1. Hydrodynamic Model

In this paper, FVCOM is used to simulate the hydrodynamic field. The FVCOM [6] is a finite-volume coastal ocean numerical model jointly developed by the School for Marine Science and Technology of the University of Massachusetts—Dartmouth and the Woods Hole Oceanographic Society, under the leadership of Dr. Changsheng Chen. The model effectively combines the advantages of the finite difference method and the finite element method. The unstructured mesh is used in the horizontal direction so that part of the terrain can be refined freely as needed. The generalized terrain-following coordinate is used in the vertical direction to better characterize complex irregular coastlines and topographies in estuaries and shelf areas. The model uses the finite volume method to numerically discretize the equation and adopts the mode-splitting algorithm to calculate the mean water level and vertical current velocity with the outer mode and to calculate physical quantities, such as temperature and salinity, with the inner mode, thus improving its calculation efficiency. The original governing equations of FVCOM mainly include the momentum equation, mass-continuity equation, and temperature, salinity, and density equations.

### 2.2. Wave Model

In this paper, SWAN is used to calculate the wave field. The SWAN [7] is a third-generation shallow-sea wave numerical model. The model adopts the spectral balance equation based on the Euler approximation and the linear stochastic surface gravity wave theory. It can simulate wave refraction, reflection, and wave shoaling caused by water-depth changes during wave generation and wave propagation and can describe the evolution of waves in nearshore areas.

### 2.3. Suspended-Sediment-Transport Model

In real estuaries and coasts, sediment movement is simulated by the bottom reference concentration, diffusion coefficient, and sediment-settling velocity in the vertical distribution model of suspended silty sediment content in response to the combined actions of waves and currents. The governing equation is a convection–diffusion equation, including a source and sink term, and the calculation method refers to the simulation process used by Ji [30]:

$$\frac{\partial c_i D}{\partial t} + \frac{\partial u c_i D}{\partial x} + \frac{\partial v c_i D}{\partial y} + \frac{\partial (w - w_{si}) c_i}{\partial \varsigma} = D \frac{\partial}{\partial x}\left(A_h \frac{\partial c_i}{\partial y}\right) + \frac{1}{D}\frac{\partial}{\partial \varsigma}\left(K_{h,s}\frac{\partial c_i}{\partial \varsigma}\right) + DS_i \quad (1)$$

where $i$ denotes the $i$th sediment component (since a single-component model is used in this paper, $i$ is 1), $c_i$ denotes the sediment concentration of the $i$th sediment component, $K_{h,s}$ denotes the vertical diffusion coefficient, $A_h$ denotes the horizontal diffusion coefficient, $w_{s,i}$ denotes the sediment-settling velocity of the $i$th sediment component, and $S_i$ denotes the source and sink term.

#### 2.3.1. Sediment-Settling Velocity

Considering that increases in sediment content hinder the sediment-settling velocity, when the median particle size of the sediment is greater than and not greater than 100 μm, respectively, the settling-velocity-hindrance formulas proposed by Slaa et al. [23] are used:

$$\begin{cases} w_s = w_{s,o}(1 - c_v)^n \\ n = 4.4\left(D_{50,ref}/D_{50}\right)^{0.2} \end{cases} D_{50} > 100 \text{ μm} \quad (2)$$

$$w_s = w_{s,o} \frac{\left(1 - \frac{c_v}{\varphi_{s,struct}}\right)^m (1 - c_v)}{\left(1 - \frac{c_v}{\varphi_{max}}\right)^{-2.5\varphi_{max}}} \quad D_{50} \leq 100 \text{ μm} \tag{3}$$

where $w_s$ denotes the sediment settling velocity in muddy water, $w_{s,o}$ denotes the sediment settling velocity in clear water, $D_{50,ref} = 200$ μm ($D_{50}$ is the median particle size of the sediment), $\varphi_{s,struct}$ denotes the structure density with a value of 0.5, $\varphi_{s,max}$ denotes the maximum density with a value of 0.65, and $m$ denotes the nonlinear effect of the wake on the settling velocity, with a value between 1 and 2.

Experiments show that the combined action of sediment concentration and gradation has an impact on the mean settling velocity, and higher sediment concentrations and gradations that that represents strong sorting ability hinder the settling velocity to a greater extent [31]. If the sediment gradation results are not taken into account in an overestimation of sediment-settling velocity, the model may underestimate the SSC and sediment transport in the water body. Therefore, in this paper, the traditional settling-velocity-hindrance formula (hereafter referred to as settling-velocity Formula (3)) and the settling-velocity formula considering gradation [31] (hereafter referred to as settling-velocity Formula (6)) are adopted for comparative calculation to consider the effect of sediment gradation on sediment-settling velocity:

$$PD \propto f[\lg(\phi_s \cdot \rho_s), \lambda] \tag{4}$$

$$\lambda = \frac{d_{90}}{d_{10}} / \frac{\sqrt{d_{25} \cdot d_{75}}}{d_{50}} \tag{5}$$

$$\frac{w_s}{w_0} = \frac{(1 - \phi_s/\phi_{s,struct})^m (1 - \phi_s)}{(1 - \phi_s/\phi_{s,max})^{-2.5\phi_{s,max}}} \cdot PD \tag{6}$$

$$PD = -0.29\left(\lambda^{0.2} - 1\right)\lg\phi_s + 1.44EXP(-\lambda) + 0.47 \tag{7}$$

where $PD$ is the gradation influence coefficient, $\lambda$ is a gradation parameter describing the gradation, and $\phi_s$ and $\rho_s$ and are the sediment-volume concentration and sediment density, respectively ($\rho_s = 2650$ kg/m$^3$).

### 2.3.2. Bottom Reference Concentration

In this study, to describe the sediment exchange between the suspended load and the bed surface, a computational simulation was carried out in the form of source and sink terms, and the formulas proposed by van Rijn. [32] and Lesser et al. [2] were used in the computation process. The height of the bed-surface reference point can be expressed as:

$$z_{ref} = max(0.5k_{s,c,r}, 0.5k_{s,w,r}, 0.01m) \tag{8}$$

The current-related bottom-roughness height is:

$$k_{s,c,r} = f_{cs}D_{50}\{85 - 65tanh[0.015(\psi - 150)]\} \tag{9}$$

$$\psi = \frac{u_b^2 + u_c^2}{(s - 1)gD_{50}} \tag{10}$$

where $f_{cs}$ is the correction factor for coarse-grained sediment, $f_{cs} = (0.0005/D_{50})^{1.5}$, and when $D_{50} < 0.5$ mm, $f_{cs}$ is 1. Furthermore, $\psi$ is the current correction coefficient, $u_b$ and $u_c$ are the bottom-current velocity and the vertical mean velocity, respectively, $k_{s,c,r}$ is in the range [0.00064, 0.075], and $k_{s,w,r}$ is the wave-related bottom-roughness height, which is considered equal to the sediment-ripple height.

The sediment concentration at the reference height is calculated according to the following formula [33]:

$$c\left(z_{ref}\right) = \max\left(\beta\eta\rho_s \frac{D_{50}}{z_{ref}} \frac{S^{1.5}}{D_*^{0.3}}, 0.05\eta\rho_s\right) \tag{11}$$

2.3.3. Sediment-Diffusion Coefficient

When there is an uneven distribution of sediment concentration in a water body, concentration stratification is formed, and the concentration gradient has an inhibitory effect on the turbulence of the water body, thereby inhibiting the diffusion of sediment. For the vertical diffusion coefficient, Yang [34] proposed a diffusion coefficient of the combined wave–current action considering the stratification effect:

$$\varepsilon_w = \varphi_d \frac{w_s l_w}{2sinh^{-1}\left(\frac{w_s}{2w_{mw}}\right)} \tag{12}$$

where $\varphi_d$ is the diffusion-correction coefficient, $\varphi_d = 1 - S$, $S$ is the inhibition rate of sediment diffusion caused by the stratification effect, where the value of $S$ is fitted based on previous experimental data ($S$ is 1 when the stratification effect is not considered), $w_{mw}$ is the mixed wave velocity, and $l_w$ is the mixed wave length. The $S$ is calculated as

$$S = -0.9\exp\left(-\frac{C\dot{v}(z)}{R_1}\right) + 0.9 \tag{13}$$

$$R_1 = \frac{1}{D_{sand}}\left[4.5 \times 10^{-8} + 2.28 \times 10^{-9}\exp\left(\frac{D_{50}/D_{sand} - 0.58}{0.118}\right)\right] \tag{14}$$

where $C\dot{v}(z)$ is the SSC gradient, $C\dot{v}(z) = -(dCv/dz)$, $Cv$ is the volumetric sediment content, $R_1$ is the empirical coefficient related to particle size, and $D_{sand}$ = 62 μm.

However, this factor only applies when the wave is not broken. After the wave is broken, the violent turbulence of the water body causes the water layers to mix with each other, the sediment-concentration gradient becomes significantly less steep, which has a significant impact on the diffusion of sediment, and there is essentially no stratification effect. When $H_s/h > 0.4$, the wave-related diffusion coefficient during wave breaking is calculated by the model proposed by van Rijn [32].

Inside the wave-boundary layer ($z < \delta_s$):

$$\varepsilon_w = 0.018\gamma_{br}\beta_w\delta_s u_b \tag{15}$$

$$\beta_w = 1 + 2(w_s/u_{*,w}) \tag{16}$$

and inside the upper water body ($z > 0.5h$):

$$\varepsilon_w = \min\left(0.05, \frac{0.035\gamma_{br}hH_s}{T}\right) \tag{17}$$

where $\gamma_{br}$ is the wave-breaking amplification factor, $\gamma_{br} = 1 + (H_s/h - 0.4)^{0.5}$, $\delta_s = 2\gamma_{br}\delta_w$ is the thickness of the boundary layer, and $u_{*,w}$ is the wave-related bottom shear velocity. The current-dependent diffusion coefficient $\varepsilon_c$ can be set as the value of the vertical eddy-viscosity coefficient calculated in FVCOM.

2.4. Model Coupling

For the three-dimensional coupled wave–current–sediment model, the coupling process can be briefly summarized as follows: The FVCOM hydrodynamic model and the SWAN wave model realize real-time exchange between the calculation elements through the MCT coupler, and the hydrodynamic model converts the water level into vertical

current velocity. The hydrodynamic model FVCOM and the wave model SWAN transfer the calculated three-dimensional current field data and wave elements to the sediment model and calculate the suspended-load-scour flux, suspended-load-siltation flux, and bed-load-transport rate through the sediment model, thereby realizing data transfer between the dynamically coupled wave–hydrodynamic model and the sediment model.

### 3. Study Area and Model Settings

#### 3.1. Study Area

Weifang Port (Figure 1) is on the south bank of Laizhou Bay. Three port areas fall within its jurisdiction, namely, the eastern, central, and western port areas. The research area of this paper is the central port area of Weifang Port, which is the main port area. To verify the rationality of the established three-dimensional coupled wave–current–sediment model, the hydrodynamic conditions and suspended-sediment conditions of Weifang Port in response to the actions of wind and waves were simulated.

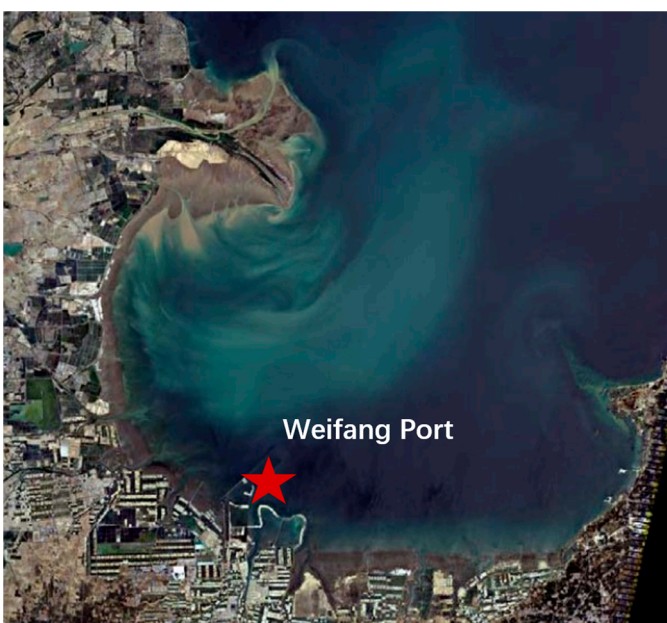

**Figure 1.** Satellite-remote-sensing map of the sea area near Weifang Port.

#### 3.2. Model Settings

The topography and water-depth data of the calculation area were the measured data of Weifang Port from 2003, and the tidal-current field and suspended-sediment verification data were the full-tide hydrological data from six hydrological stations from 10–11 November 2003. The wind-field data were derived from the ERA5 wind-field-reanalysis product. The ERA5 is a fifth-generation high-resolution reanalysis dataset developed by the European Centre for Medium-Range Weather Forecasts by assimilating multisource observational data. This dataset combines current measured data with previous forecast results every 12 h to obtain accurate atmospheric forecast results. At present, users can obtain the hourly wind-field data from 1979 to the present, with a temporal resolution of 1 h and a spatial resolution of $0.25° \times 0.25°$. In this paper, the ERA5 data for a wind-field height 10 m above the Earth's surface were selected as the wave-drive conditions of the SWAN model. The particle-size-distribution values were obtained by measuring and analyzing the sediment samples collected in the sea area near Weifang Port using the Malvern 3000 particle-size analyzer (Figure 2). The median particle size of the sediment was 0.066 mm, and the gradation parameter of the sediment in Weifang Port was calculated as 4, according to settling-velocity Formula (6).

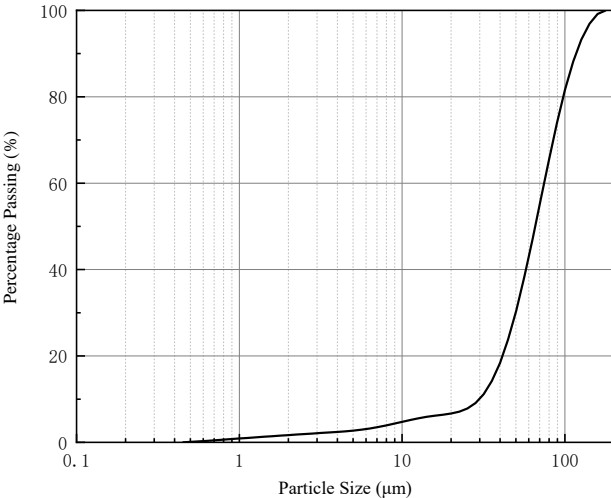

**Figure 2.** Sediment-particle-size distribution in the sea area near the Weifang Port.

To ensure the accurate calculation of the tidal-current field in the study area, the method of nesting large and small grids was adopted, and the large and small models both used unstructured triangular grids. The large model included the entire Bohai Sea, and the grid was refined in the sea area near the Weifang Port. The calculation range was from 37°1′ N–40°52′ N to 117°32′ E–122°13′ E, the mesh scale was between 3500 m and 4000 m, and the grid number was 12,526. The grid and water-depth data are shown in Figure 3. The small model mainly included the sea area near the project area, and the calculation range was from 37°5′ N–38°32′ N to 118°50′ E–119°32′ E, the mesh scale was between 20 m and 1500 m, and the grid number was 14,763. The grid and water-depth data are shown in Figure 4. The model was vertically divided into 10 layers. The wetting and drying algorithms were used, and the minimum water depth was set to 0.02 m.

The water-level-boundary conditions were used for the open boundaries of the large and small models. The open-boundary water-level data of the large model were derived from the MIKE21 global-tide-forecasting system, and the ERA5 wind field was used as the wave-driving condition for the large model. The open-boundary water-level data and wave-boundary conditions of the small model were extracted from the calculation results of the large model.

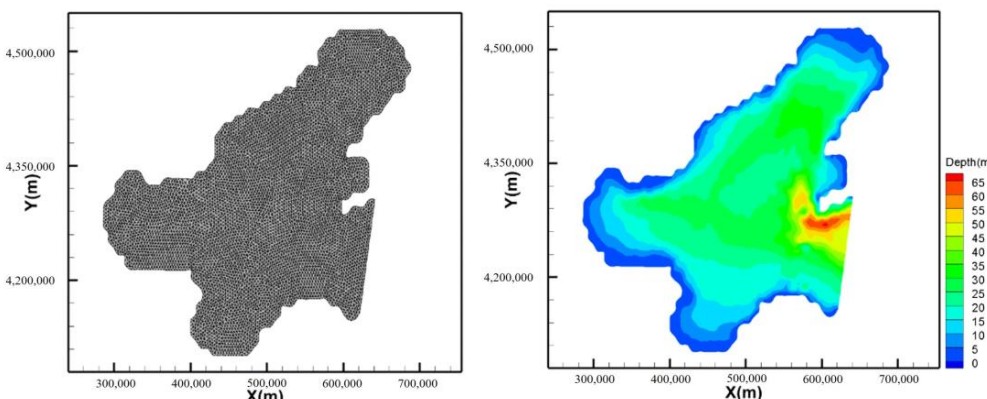

**Figure 3.** Large model and water depth.

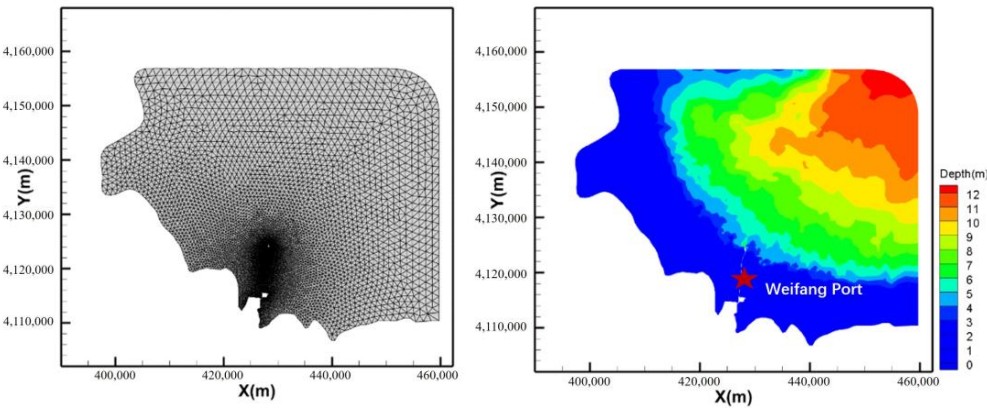

**Figure 4.** Small model and water depth.

## 4. Model Results and Analysis

### 4.1. Tide Elevation and Tidal-Current Verification

The measured hydrological data used in this study were from 10 to 11 November 2003. They included the tidal elevation, tidal-current velocity, tidal-current direction, and SSC. The locations and specific coordinates of the stations are shown in Figure 5 and Table 1, respectively.

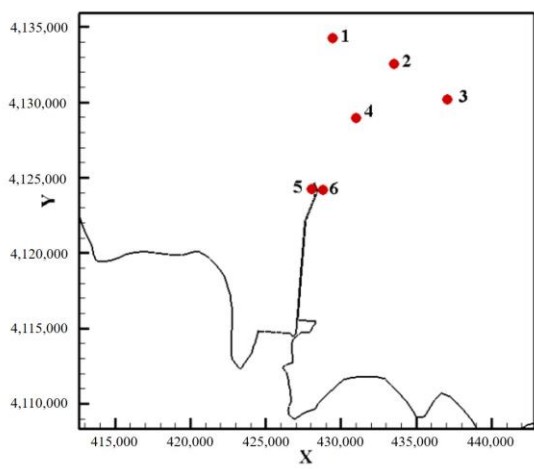

**Figure 5.** Locations of the stations.

**Table 1.** Coordinates of the stations.

| Station Number | Beijing54 | | WGS84 | |
|:---:|:---:|:---:|:---:|:---:|
| | x | y | E | N |
| 1 | 429,431.5 | 4,134,252 | 119.2 | 37.34 |
| 2 | 433,521.9 | 4,132,544 | 119.25 | 37.32 |
| 3 | 437,045.2 | 4,130,220 | 119.29 | 37.30 |
| 4 | 431,003.2 | 4,128,988 | 119.22 | 37.29 |
| 5 | 428,060.5 | 4,124,240 | 119.19 | 37.25 |
| 6 | 428,809.7 | 4,124,215 | 119.20 | 37.25 |

Figure 6 shows the comparison between the simulated tide levels (using settling-velocity Formula (6)) and the measured tide levels from 12:00 on 10 November 2003 to 20:00 on 11 November 2003 at Station 1 in the sea area near Weifang Port. The verification

results were quite consistent. Figures 7 and 8 show the stratified verification results of the tidal-current velocity and direction at each station in the sea area near Weifang Port from 14:00 on 10 November 2003 to 17:00 on 11 November 2003. The simulated current velocities and directions of Station 1 to Station 4 were generally consistent with the measured results. The near-bottom-current velocities simulated by Stations 5 and 6 were slightly slower than the values measured at certain time points, and the simulated current directions also deviated somewhat from the measured data. This is probably because Stations 5 and 6 were in the vicinity of a structure, so the current there is greatly affected by the terrain and boundaries. Overall, the simulated current velocities and directions at the six stations above were close to the measured current velocities and directions in the continuous diachronic change process, and the simulated and measured phases were generally consistent. The three-dimensional water-and-sediment model used in this paper reasonably reflects the hydrodynamic patterns of the sea area, so the model can be used for suspended-sediment simulations.

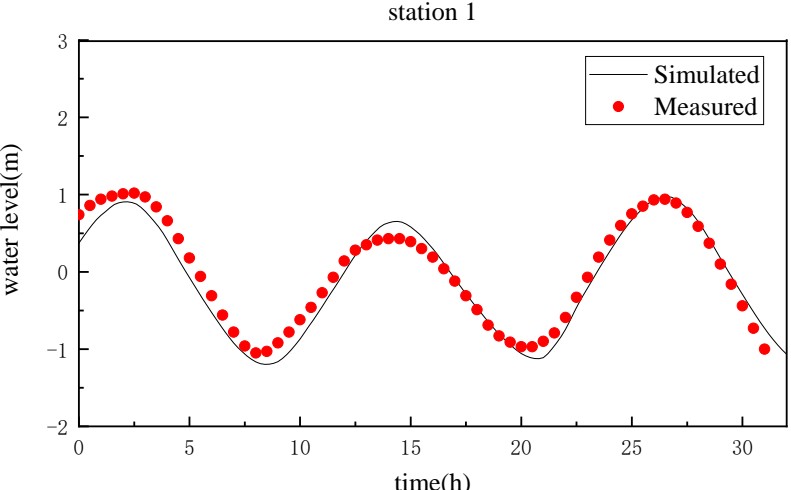

**Figure 6.** Tide-level verification at Station 1.

### 4.2. Suspended Sediment Concentration Verification

This paper uses the measured data for the suspended sediment in the sea area near Weifang Port from 10–11 November 2003 to validate the SSCs in the bottom and surface water bodies. Figures 9 and 10 show the suspended-sediment-verification conditions from Stations 1 to 6 (using settling-velocity Formula (6)). The sediment concentration in the surface water body was relatively low, while the sediment concentration in the bottom water body was relatively high. The SSC of each layer fluctuated regularly with time, and the fluctuation amplitude also increased with the water depth. The SSCs in the surface layer of Station 4, the bottom layer of Station 5, and the bottom layer of Station 6 were underestimated in a few time periods. The SSC was related to the bottom-current velocity. If the numerical model underestimated the current speed near the seabed, the sediment was not easy to start, resulting in lower SSC in the water bodies. And this chain reaction had a lag in time, that is, the lower simulated SSCs generally occurred after the simulated current speed was small. According to the previous current-velocity and direction0verification figures, the simulated current velocities were slower than the measured values in the surface layer of Station 4 at 10–15 h, in the bottom layer of Station 5 at 0–5 h, and in the bottom layer of Station 6 at 12–18 h, which caused the underestimation of the SSCs at these time points. Overall, the trend and magnitude of the measured and simulated values at most of the stations were generally the same, so the verification results were good, indicating that the model that uses settling-velocity Formula (6) can effectively simulate the actual sediment movement in the sea area.

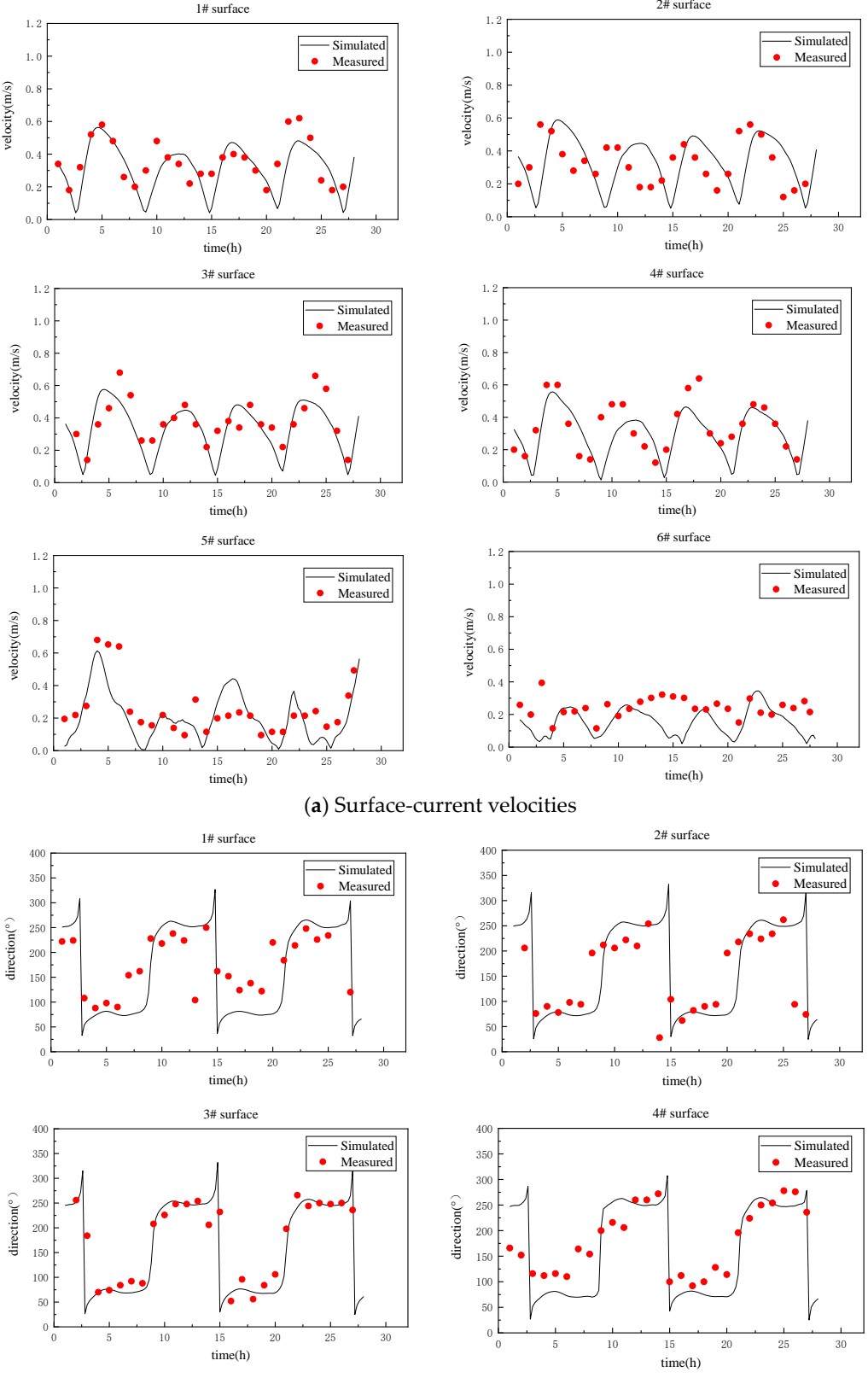

(**a**) Surface-current velocities

**Figure 7.** *Cont.*

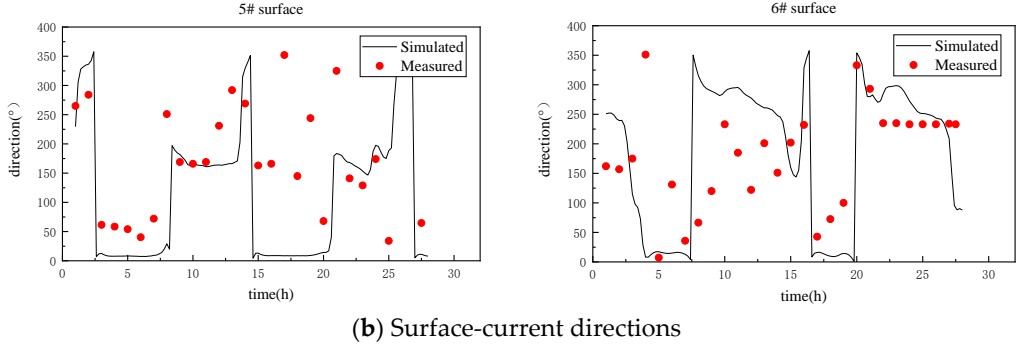

(**b**) Surface-current directions

**Figure 7.** Verification of surface-tidal-current velocities and directions.

(**a**) Bottom-current velocities

**Figure 8.** *Cont.*

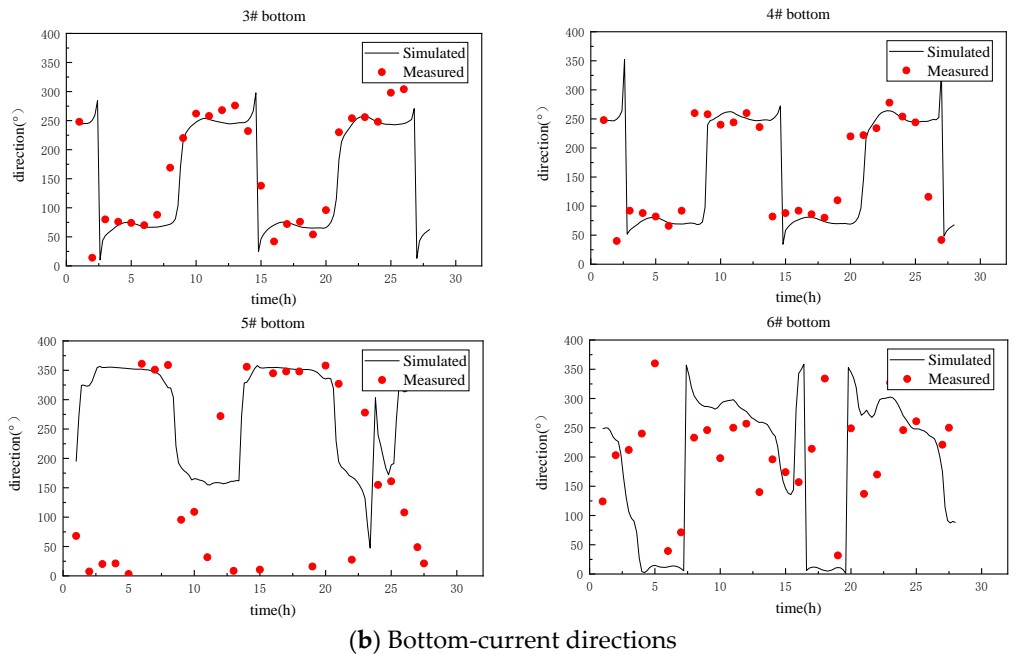

(**b**) Bottom-current directions

**Figure 8.** Verification of bottom-tidal-current velocities and directions.

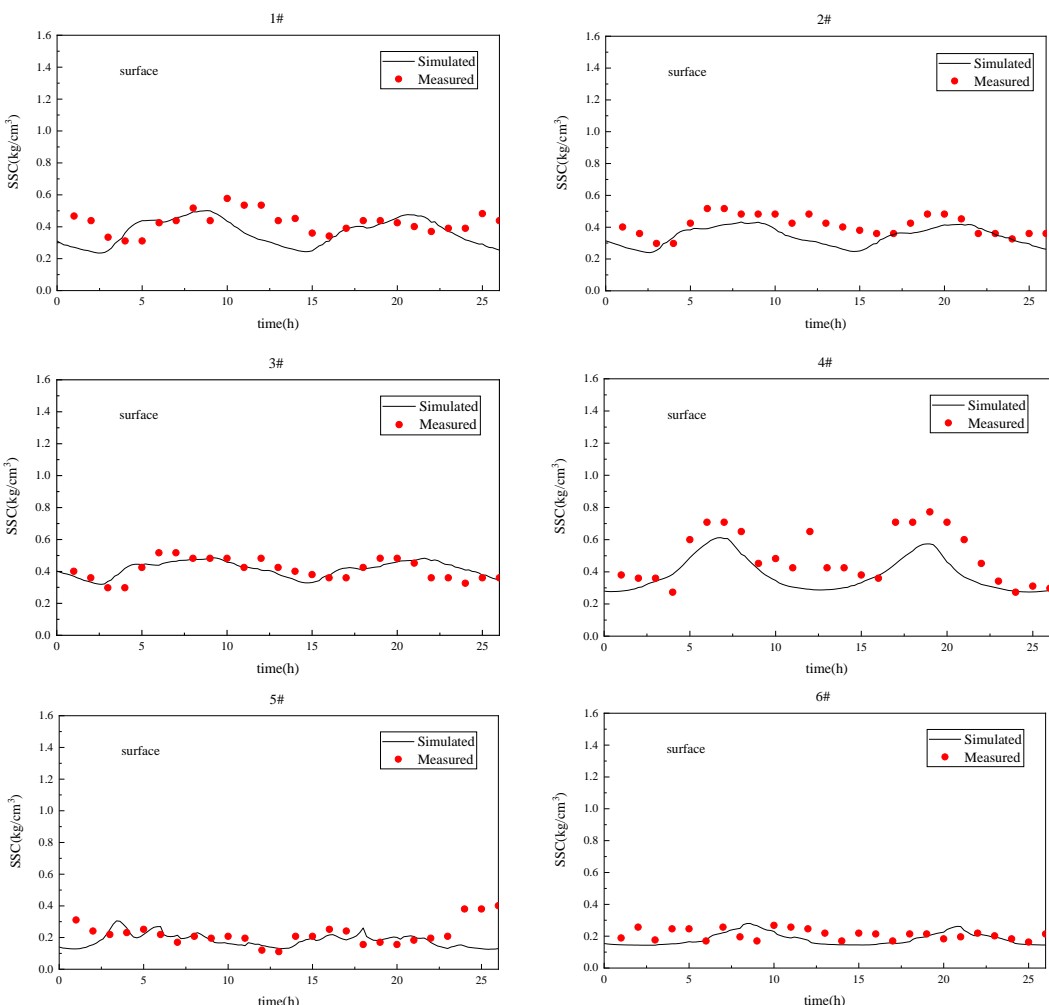

**Figure 9.** Validation of suspended-sediment concentrations in the surface layers of the stations.

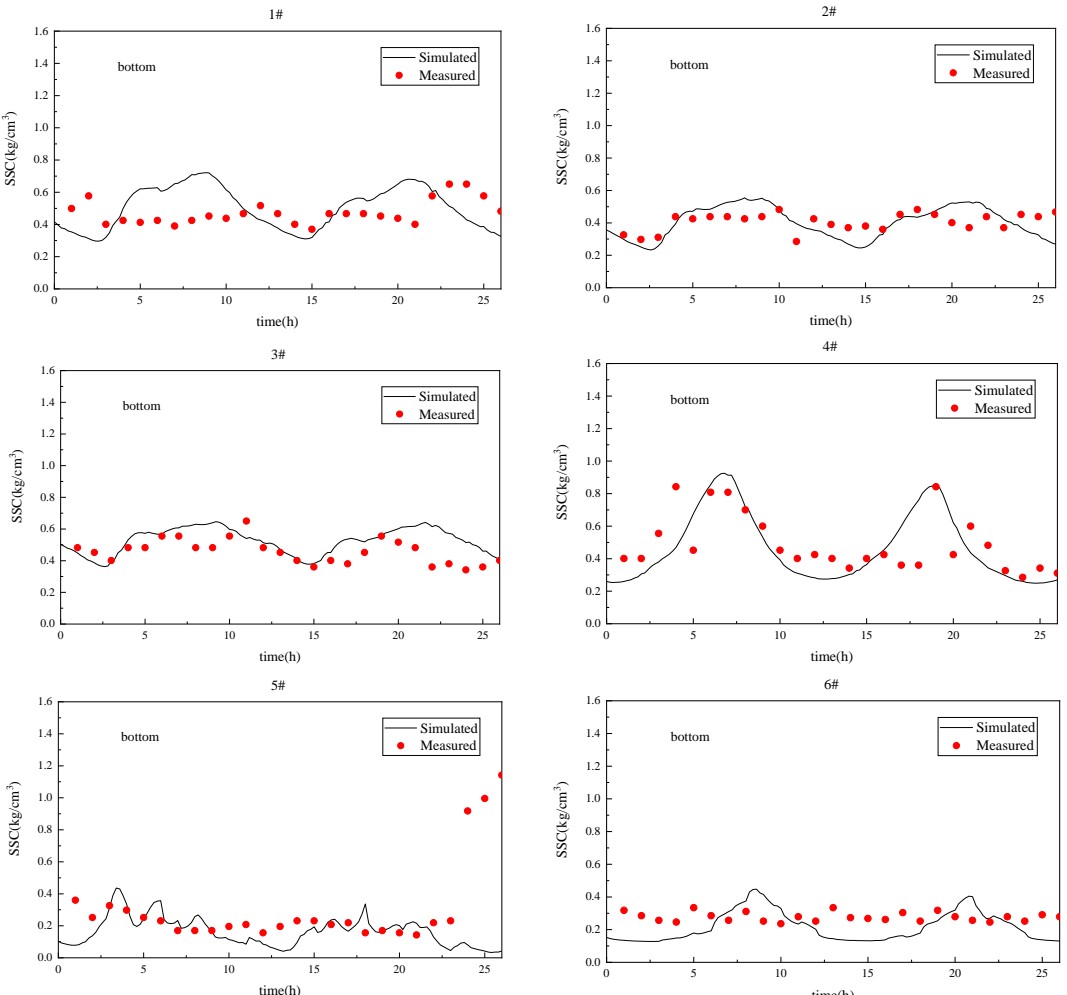

**Figure 10.** Validation of suspended-sediment concentrations in the bottom layers of the stations.

*4.3. Sediment-Content Comparison*

Settling-velocity Formula (3) was used to simulate the sediment in the sea area near Weifang Port during the same time period using the same parameter settings as above. Taking Station 4 as an example, the simulated SSCs of the surface and bottom layers before and after the correction are shown in Figure 11. The simulated SSCs after the correction were higher than those before the correction. The maximum current velocity reached 0.5 m/s between 5–10 h and 17–22 h, and the differences between the simulated SSCs before and after the correction were larger during this period. After introducing the modified sediment settling-velocity formula, the overall sediment velocity was lower than that simulated by the original settling-velocity formula. When the current velocity was high, a significant amount of sediment was suspended, the suspended sediment settled less easily, and the SSC in the water body increased and fluctuated more.

The values of mean relative error (*MRE*) before and after the correction were used to measure the effect of the gradation on the distribution of the suspended sediment. The root mean square error (*RMSE*) represents the sample standard deviation of the differences between the predicted values and the experimental values. The smaller the *MRE* and the *RMSE*, the better the predicted values fit the experimental values. The *MRE* and *RMSE* are calculated as:

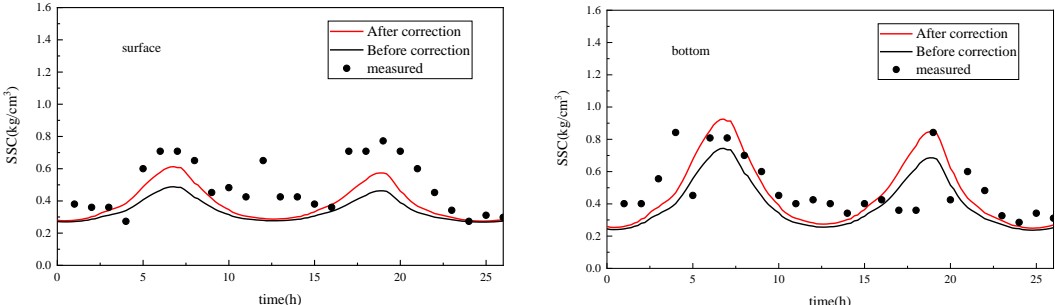

**Figure 11.** Comparison of suspended-sediment concentrations at Station 4 before and after the correction.

$$MRE = \frac{1}{n} \sum_{i=1}^{n} \frac{\left| C_{i,After\ correction} - C_{i,Before\ correction} \right|}{C_{i,Before\ correction}} \tag{18}$$

$$RMSE = \sqrt{\frac{1}{m} \sum_{i=1}^{m} C_{i,Measured} - C_{i,Simulated}} \tag{19}$$

where $n$ is the number of vertical position points calculated by the model, $m$ is the number of stations, $C_{i,Measured}$ is the measured SSC, and $C_{i,Simulated}$ is the simulated SSC.

According to Formulas (18) and (19), the deviations of the simulated SSCs from the measured SSCs were calculated before and after the correction, respectively. For the SSCs in the surface layers, the *MRE* values before and after the correction were 29% and 22%, respectively, and the *RMSE* values before and after the correction were 0.19 kg/m³ and 0.14 kg/m³. For the SSCs in the bottom layers, the *MRE* values before and after the correction were 22% and 15%, respectively, and the *RMSE* values before and after the correction were 0.19 kg/m³ and 0.13 kg/m³. The calculation results show that the results calculated by the modified model were more accurate.

Figures 12–14 show the wave fields, current fields, and SSC fields, respectively, in the surface and bottom layers during a surge before and after the correction. The analysis of the SSC field maps shows that the SSCs in the open sea were relatively small (mostly less than 0.5 kg/m³ in both the surface and the bottom layers) and that in the nearshore area, due to wave shoaling and breaking, the SSCs exceeded 2 kg/m³. In the nearshore area, the SSCs were affected by the current velocity, and the SSCs were higher at locations where the velocity was high or changed drastically. The current field mainly pointed from east to west. Due to the occlusion of the structure, the SSCs were low on the west side of the structure and high on the east side of the structure. Comparing Figures 13 and 14, the nearshore SSCs calculated by the settling-velocity formula proposed in this paper were higher than those calculated by the original settling-velocity formula. Specifically, the nearshore SSCs in the surface and bottom layers calculated by the settling-velocity formula proposed in this paper were approximately 1 kg/m³ and 2 kg/m³ higher, respectively than those calculated by the original settling-velocity formula. In practical engineering applications, the SSCs calculated by the settling-velocity formula proposed in this paper will be even higher, so a construction scheme with a higher safety factor is recommended for the study area.

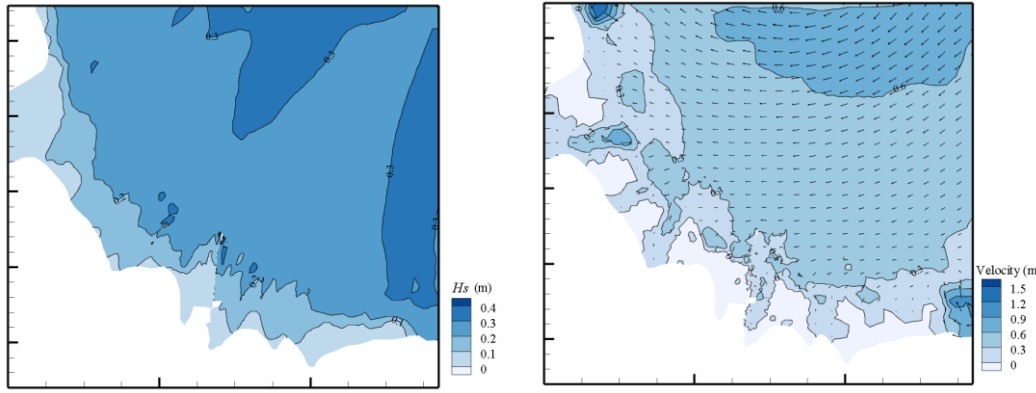

**Figure 12.** The wave field and current field during a surge.

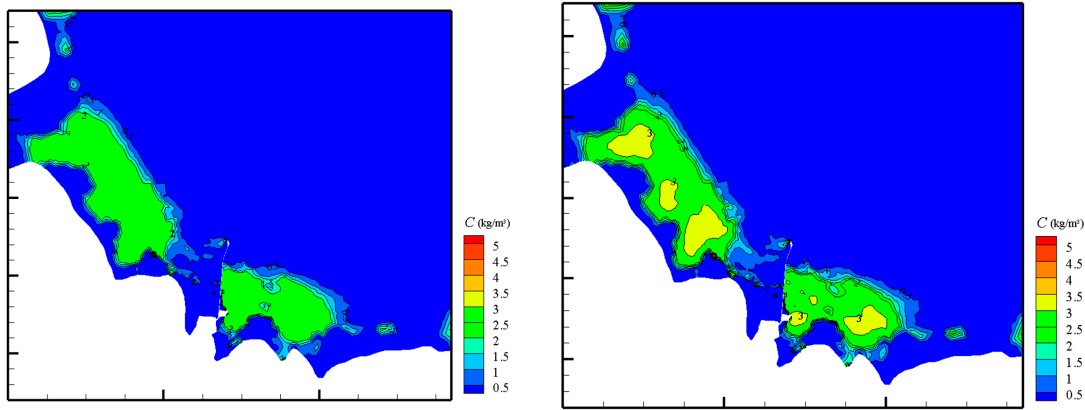

**Figure 13.** Suspended-sediment-concentration fields in the surface and bottom layers (before correction).

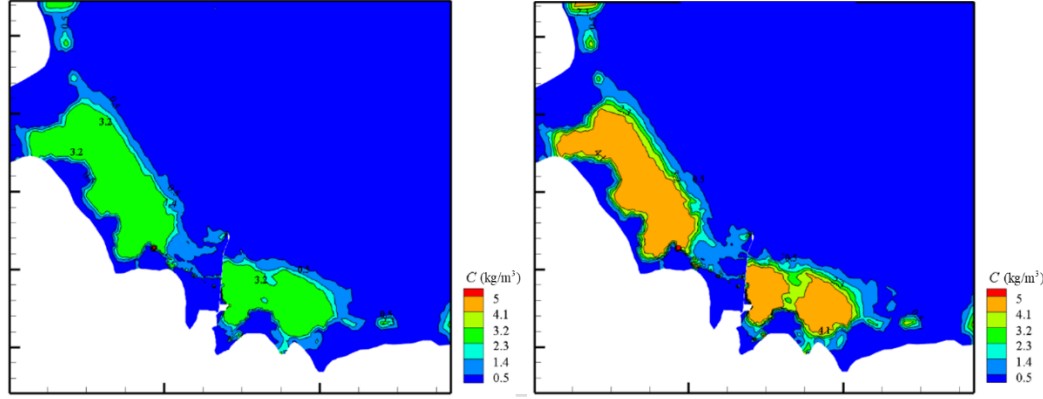

**Figure 14.** Suspended concentration fields in the surface and bottom layers (after correction).

## 5. Conclusions

This study introduced a sediment-settling-velocity formula that considers gradation in the three-dimensional FVCOM-SWAN coupled water-and-sediment-movement model and simulates the suspended-sediment movement in the sea area near Weifang Port with the modified single-component model. We drew the following conclusions:

(1) After introducing settling-velocity Formula (6), the overall settling velocity of the sediment decreased. The higher the sediment concentration is, the more the settling velocity is tempered. The sediment in the bottom water body was more highly concentrated than that in the surface water body. The SSC in the bottom layer was

high and fluctuated more. After the introduction of settling-velocity Formula (6), the model fitted the measured data better. Hence, the model can effectively describe the sediment-movement process in the sea area near Weifang Port.

(2)  The SSCs simulated by settling-velocity Formula (6) were higher than those simulated by settling-velocity Formula (3), and the SSCs simulated by the two formulas differed more when the current velocity was faster. With settling-velocity Formula (6), the overall settling velocity of the sediment was slower than that simulated by settling-velocity Formula (3). When the current velocity was high, more sediment was suspended, the suspended sediment settled less easily, and the SSCs in the water body increased and fluctuated more.

(3)  For the SSC field in the sea area of Weifang Port, the nearshore SSCs calculated by settling-velocity Formula (6) were higher than those calculated by settling-velocity Formula (3). Specifically, the nearshore SSCs in the surface and bottom layers calculated by settling-velocity Formula (6) were approximately 1 kg/m$^3$ and 2 kg/m$^3$ higher, respectively than those calculated by settling-velocity Formula (3). In practical engineering applications, the SSCs calculated by a settling-velocity formula considering gradation will be even higher, so a construction scheme with a higher safety factor is recommended for the study area.

**Author Contributions:** J.Q., conceptualization, methodology, investigation, and formal analysis. Y.J., investigation, formal analysis, validation, and writing—original draft preparation. C.C., investigation and data curation. J.Z., writing—review and funding acquisition. All authors have read and agreed to the published version of the manuscript.

**Funding:** This study was supported by the National Natural Science Foundation of China (grant nos. U1906231), and the Open Funds of State Key Laboratory of Hydraulic Engineering Simulation and Safety of China (grant no. HESS-2221).

**Data Availability Statement:** Not applicable.

**Acknowledgments:** The authors also gratefully acknowledge the comments and suggestions of the anonymous reviewers.

**Conflicts of Interest:** The authors declare no conflict of interest.

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
