# Peer review of "Numerical Simulation of Tidal Current and Sediment Movement in the Sea Area near Weifang Port"

_water, doi:10.3390/w15142516_

Round 1

Reviewer 1 Report

This paper introduced a graduation parameter into the sediment settling velocity formula. Further, the new fall velocity formula was incorporated into a coupled wave–current–sediment numerical model. Observed hydrodynamic data and measured suspended sediment concentration were utilized to examine the performance of this model. Numerical results agreed generally well with the observed data. Applying the modified settling formula,high sediment concentration was well described near the seabed on silty coasts.

Overall, this paper is reasonably organized. The modified numerical model has revealed some new features of sediment transport on silty coasts. Additional revisions are as follows:

(1) There are some grammatical mistakes in the paper. The English writing of this manuscript should be improved.

(2) Illustrating the location of Weifang Port in Figure 1 and Figure 4.

(3) Try to elucidate the reason for modifying the formula of settling velocity in detail.

(4) Chinese characters are shown in Figure 9, Figure 10 and Figure 11.

(5) In Figure 11, simulated bottom sediment concentration was significantly improved by applying the new settling velocity formula. However, sediment concentration in the sea surface was slightly improved using the modified model. Specify the primary reasons.

(6) In Section 3.1, ‘tide level’ should be replaced to ‘tidal elevation’.

(7) Figure 8a indicates that the numerical model has underestimated the current speed near the seabed. Try to discuss the impacts on predicting sediment concentration near the seabed.

There are some grammatical mistakes in the paper.

Reviewer 2 Report

The paper titled "Numerical Simulation of Tidal Current and Sediment Movement in the Sea Area near Weifang Port" mainly introduces a sediment settling velocity formula that incorporates gradation into the three-dimensional FVCOM-SWAN coupled water and sediment movement model. It simulates the suspended sediment movement in the sea area near Weifang Port using the modified single-component model. This paper is significant; however, there are several issues that need improvement before its publication.

  1. The formula's corner mark on line 141 in this paper is written incorrectly. Please correct it.

  2. In this paper, ERA5 data with a height of 10 m above the Earth's surface is selected as the components of the south and west wind. Please provide a detailed explanation for this statement.

  3. Can the selected grid scale in this paper be appropriately expanded to reduce the total number of grids and the computational load? Will this expansion affect the accuracy of the results?

  4. Is it possible to apply the coupled model to predict sediment suspension mass movement and make corresponding decisions and management based on the prediction results?

  5. It is recommended to use the sensitivity analysis method in this paper to observe the changes in the output results of the model by varying the values of each parameter in the coupled model. This will allow for an evaluation of the accuracy and applicability of this coupled model.

  6. In this paper, only the influence of wind fields and waves on the sediment settling velocity is studied. Should the influence of ocean currents, surrounding land environmental factors, and physical and chemical parameters of water bodies on sediment movement be considered?

Minor editing of English language required

Round 2

Reviewer 2 Report

The author has made thorough revisions, and the suggestions can be accepted.

Minor editing of English language required.